# FEM Modeling Strategies: Application to Mechanical and Dielectric Sensitivities of Love Wave Devices in Liquid Medium

**DOI:** 10.3390/s24102976

**Published:** 2024-05-08

**Authors:** Maxence Rube, Ollivier Tamarin, Asawari Choudhari, Martine Sebeloue, Dominique Rebiere, Corinne Dejous

**Affiliations:** 1UMR 228 Espace-Dev, University of French Guiana, F-97300 Cayenne, France; ollivier.tamarin@univ-guyane.fr (O.T.); asawari.choudhari@univ-guyane.fr (A.C.); martine.sebeloue@univ-guyane.fr (M.S.); 2Laboratoire de l’Intégration du Matériau au Système, University of Bordeaux, CNRS, Bordeaux INP, UMR 5218, F-33405 Talence, France; dominique.rebiere@u-bordeaux.fr (D.R.); corinne.dejous@ims-bordeaux.fr (C.D.)

**Keywords:** finite element modeling, surface acoustic wave biosensor, Love Waves, COMSOL

## Abstract

This paper presents an extended work on the Finite Element Method (FEM) simulation of Love Wave (LW) sensors in a liquid medium. Two models are proposed to simulate the multiphysical response of the sensor. Both are extensively described in terms of principle, composition and behavior, making their applications easily reproducible by the sensor community. The first model is a Representative Volume Element (RVE) simulating the transducer and the second focuses on the sensor’s longitudinal (OXZ) cut which simulates the multiphysical responses of the device. Sensitivity of the LW device to variations in the rheological and dielectric properties of liquids is estimated and then compared to a large set of measurements issued from LW sensors presenting different technological characteristics. This integral approach allows for a deeper insight into the multiphysical behavior of the LW sensor. This article also explores the advantages and drawbacks of each model. Both are in good accordance with the measurements and could be used for various applications, for which a non-exhaustive list is proposed in the conclusion.

## 1. Introduction

Finite Element Modeling (FEM) is inextricably linked with materials and mechanics studies and has gained a lot of attention and applications in electronic systems. Other well-known conventional methods can be easier and use less calculation resources, such as those based on the Coupled Mode Theory (CMT), delta method or analytical equations of the acoustic waves propagation [1,2,3]. However, most strategies assume extreme simplifications of the device and its physical behavior, for example, when considering the piezoelectric substrate as a semi-infinite medium [4]. Nonetheless, such a modeling approach remains well documented and gives good results as a first approximation.

FEM of acoustic devices is receiving increasing attention by the sensor community. However, the general FEM strategy applied to electroacoustic sensors, as presented in the literature, often uses even more simplified models, such as reducing the size of the transducers or of the device, achieving either similar or less accurate results compared to common methods. For example, Kovacs et al. [5] calculated the optimum sensitivity of a Love Wave (LW) sensor with an analytical model in 1992, while Rocha-Gaso et al. [6] were the first to reproduce this application using FEM in 2010, with an extremely simplified model and a time-consuming method.

Our work proposes two quite novel models based on FEM to investigate the multi-physical response of an LW sensor delay-line in a liquid medium. Although dedicated solutions to ensure a purely acoustic response of LW devices were intensively investigated, this work consists of going further than the typical assumptions made in [7,8]. Indeed, Surface Acoustic Wave (SAW) devices can be used as capacitive and acoustic sensors with electrical and acoustic-related sensitivities [9]. Hence, the use of FEM could allow complete decorrelation of each physical phenomenon from the response of the LW sensor in a complex fluid. The use of those models could also be extended to design and optimization of LW devices and their instrumentation for liquid medium sensing.

This paper is structured into four sections. The Section 2 describes the generalities of LW sensors along with a short state of the art of SAW devices modeling. Then, a short description of the LW sensor used for this work and a brief timeline of the FEM method for the simulation of LW sensors are provided. The Section 3 describes the materials and methods employed. Subsequently, both models are extensively described in terms of principle, composition and behavior. Based on that, Section 4 and Section 5 compare the measurements and results of the models in liquid, respectively, for dielectric and mechanical sensing estimation. For each model, results, performances, advantages and drawbacks are discussed in a short subsection for synthesis purposes. Finally, the Section 6 proposes some of the applications of the developed models and presents further work for improvement.

## 2. Context and State of the Art

### 2.1. Love Wave Sensor in Liquid Medium—A Growing Complexity

The experimental LW sensor used in this study is a well-known delay line whose physical structure is detailed in [9]. Basically, this device consists of two Inter Digital Transducers (IDTs) with a structure similar to planar capacitors. The IDTs are located at the surface of an AT cut quartz piezoelectric substrate. When an electrical sinusoidal signal is applied to the input transducer, an electro-mechanical conversion occurs via a reverse piezoelectric effect, creating an electro-acoustic wave with shear horizontal polarization. This acoustic wave propagates around a resonance frequency “$f_{r}$” (in our case about 115 MHz) through the “acoustic path”, from the input toward the output IDT, where an acoustic to electrical conversion occurs via a direct piezoelectric effect. Finally, a thin layer of silicon dioxide, of a few micrometers thickness (“$H_{GL}$”) is deposited over the surface of the quartz substrate. This thin layer traps and guides the acoustic wave, also preventing a top liquid medium from short-circuiting the IDTs. A simplified representation of the device with a basic electrical read-out instrumentation is schematized in Figure 1.

Love Wave devices have shown promising applications for biological detection in a liquid medium, using the acoustic wave propagation to measure a mass-loading effect at the surface of the sensor [10]. Also, applications characterizing liquid mechanical properties as viscosity and density are well known [10,11]. To better understand the multiphysical aspect of the sensor, a more thorough description of the sensor power flow has to be given.

The input impedance of the LW device can be separated into two components: the electrical and the electroacoustical impedances. Basically, the geometry of the LW device, and more specifically the construction of the IDTs, gives the purely electrical impedance of the sensor. According to this, we can assume that the imaginary part of the impedance is mainly caused by the obvious capacitance, inherent to the interdigitated geometry of the IDTs. In addition, metal IDTs are assumed to exhibit electrical resistivity. Finally, the electro-acoustic transduction will cause a change in the impedance around the resonant frequency $f_{r}$ [1,12,13].

Both electrical and mechanical properties of the liquid medium interact with the sensor in different ways. In terms of sensing, a change in the electrical properties (dielectric permittivity and conductivity) of the liquid sample affects the electrical impedance of the IDTs. When considering the input port, this causes a change in the reflected wave $b_{1}$, and thus in the $S_{11}$ parameter and subsequently in the transmitted wave $b_{2}$, affecting the $S_{21}$ parameter. A change in the mechanical properties of the solution (viscosity and density of the liquid or mass loading) affects the acoustic wave propagation properties around $f_{r}$, inducing a frequency shift of $f_{r}$. This phenomenon changes the impedance around the resonance frequency $f_{r}$ and also causes a change in the celerity and the attenuation of the propagating wave $b_{2}$ thus changing both $S_{11}$ and $S_{21}$ parameters. Attenuation and phase shift are the most commonly used measurements for sensing applications. In addition, an electrical coupling also occurs because of the dielectric permittivity of the liquid, creating an “energy” conduction channel from the input to the output transducer.

To illustrate the multiphysical phenomena occurring in the sensor, an alternative methodology highlighting different energy flows was proposed in [9] and is schematically depicted in Figure 2.

Thus, as presented in Figure 2, concerning the experimental “multiphysical” behavior of the LW device, the use of the FEM model could allow us to perfectly analyze the response of the sensor for complex fluid parameters, increasing our understanding of the responses.

### 2.2. A Brief History of FEM Modeling of Love Wave Sensor

Although extremely powerful and versatile, FEM can require enormous computational resources, depending on the task. Hence, it is of extreme importance to use geometric simplification and boundary conditions to limit the number of elements in the model. However, this simplifying approach might be neither simple nor obvious. The first modelization of a Love Wave propagation using FEM was made by Lysmer et al. [14] in 1972 for an application of seismic wave observation. However, the first application to a SAW device only came with Ventura et al. [15] who introduced both BEM (Boundary Element Method) and FEM to simulate a Rayleigh wave device. This model used a frequency domain simulation of a simplified periodic structure of IDTs in two dimensions, receiving a lot of his inspiration from the COM and delta methods. Ippolito et al. [16] proposed the first 3D FEM model of a Shear Horizontal SAW device, designing an RF filter using Lithium Niobate substrate with a zinc oxide guiding layer. Ippolito’s model used both the frequency and the time domain simulation with only a pair of fingers per IDT and the results obtained were quite different from the actual measurement. An FEM approach was also employed by Rocha-Gaso et al. [6] for the evaluation of the sensitivity of a Love Wave device to mass effect. More recently, Achour et al. [17] chose the FEM approach in a frequency domain simulation for the design of an SH-SAW device and also proposed a 2D model reducing calculation needs in resources.

Although improvements are clearly being achieved among the sensors community, simplifications of FEM models considerably affect the system behavior. Most models tend to describe the response of the sensor rather than the accurate behavior of the physical sensor. This strategy considerably reduces the interest for FEM simulations in understanding the multiphysical aspects of the device.

## 3. Material and Methods

The simulations were conducted on COMSOL Multiphysics software version 5.2 for Finite Element Method Modeling, on a computer with an AMD Threadripper 2990WX processor unit, 128 GB of RAM and Nvidia QUADRO M2000 as a graphical processor unit. Both models were developed on the same system.

### 3.1. Representative Volume Element Model

The Representative Volume Element (RVE) is a modeling approach consisting in simulation of a unit cell, chosen to be either statistically or structurally representative of the global system. COM and delta models use this method intensively for SAW device simulations due to the periodicity of the IDTs [18,19,20]. Both COM and FEM can be used to better simulate the sensor response, such as in [21]. A typical representation of a SAW RVE model is presented in Figure 3.

In our case, as presented in Figure 4, the RVE model of the LW sensor consists of one IDT with one pair of double fingers. Periodic conditions are applied to each lateral edge and a low reflection boundary condition is applied to the bottom edge of the quartz substrate. The quartz substrate is modeled with two physical halves, viz. top and bottom. The top part of the substrate, close to the IDTs and guiding layer, takes into account the electro-mechanical behavior, whereas the bottom half, with weak mechanical deformations, takes into account only the electro-static physics of the sensor. Ground at zero voltage is applied on the bottom face of the model and two of four electrodes of the IDTs, in line with the split-finger electrode design. A 1 V magnitude alternating voltage is applied to the other half. Structurally, the metal IDTs are modeled using the actual thickness geometry ($H_{met}$ in Figure 3), that is, three layers successively of 40 nm of titanium, 90 nm of gold ($H_{AU}$ in Figure 3) and 10 nm of titanium. The RVE model is one wavelength wide and long, but takes into account the real thickness of the device (500 μm thick quartz substrate and ≈4 μm thick $SiO_{2}$ guiding layer—$H_{GL}$ in Figure 3).

Because of the symmetric simplifications made to the model and thanks to the use of periodic conditions, the model just needs a mesh on the Oxz plan. Each thin layer, guiding layer and IDT “layer” has correspondingly eight and five mesh elements per thickness, with a maximum elements size of 10.5 μm on the lateral edge. The first half of the substrate has a maximum element size of 5 μm, to be at least 1/8th of the wavelength. The second half of the substrate has a maximum element size of 10 μm. Simplification of the model by reducing the substrate thickness did not exhibit any computational gain, nor effect the calculated results. Therefore, the substrate was modeled with the actual sensor thickness.

The RVE model allows two kinds of simulations: eigenfrequency and/or frequency-domain. The eigenfrequency simulation gives the frequencies at which resonance and anti-resonance occur. As periodic conditions are applied, the system is equivalent to a SAW sensor with an “infinite” number of IDTs. Subsequently, the difference in the resonance and anti-resonance frequencies is near 100 Hz, while the resonance frequency $f_{r}$ is near 110 MHz. This simulation gives a resonance frequency as a result, which is exclusively sensitive to mechanical parameters. Hence, this model can be used only for the mechanical parameter variations, but with a very high precision.

In addition, the frequency domain simulation gives the impedance of the IDTs at a fixed frequency. The virtually high Q factor makes it difficult to use a frequency sweep for the estimation of a mechanical parameter on $f_{r}$. Indeed, the frequency sweep requires a tremendous number of points to obtain a frequency step small enough to finely observe the frequency shifts of the LW transduction. This minimum frequency step that affects the numerical precision will also affect the accuracy of the output, in contrast to the eigenfrequency solver results. Nonetheless, interrogating the device at different frequencies might allow us to separate electric ($f<<f_{r}$) from electro-acoustic ($f\approx f_{r}$) behavior.

When estimating the response of the sensor in the air, a volume is placed directly on the surface of the guiding layer. This volume simulates only electrostatic physics, with an electrical permittivity of 1. In cases of operation in a liquid medium, due to the low permittivity of the quartz substrate and $SiO_{2}$ guiding layer and also the capacitive aspect of the device, stable results were found for a simulated liquid thickness superior to 30 μm. Therefore, the LW RVE model considers a sample volume of thickness 50 μm with electro-static physics behavior for both the liquid and air media. This volume is meshed with a maximum element size of 5 μm, to be at least 1/8th of the wavelength.

More specifically, when estimating the simulated LW response to the electrical and mechanical parameters of a solution, the liquid is hypothesized to behave like a Newtonian fluid. Therefore, we used a rheological Kelvin–Voigt equivalent model with a purely complex shear modulus caused by the dynamic viscosity and density and no real shear modulus. Moreover, still in a liquid medium, the RVE model uses an extremely well meshed thin layer of 1 μm (included in the whole 50 μm thick modeled liquid layer), with elements every 30 nm. This well meshed strategy of 1 μm thick liquid layer is sufficient to model the mechanical dissipation of the SH component of the LW. Indeed, the theoretical penetration depth factor is ≈50 nm in water, for a density and a dynamic viscosity of 1 g/mL and 1 cP, respectively, at a frequency of 100 MHz [11]. Finally, a low-reflection boundary condition is also imposed at the top face of this liquid unit model. As previously explained, typical LW sensors use transiting electroacoustic energy as a sensitive mean, which is measured with the $S_{21}$ parameter. This transiting energy does not exist for the RVE cell, being purely resonant. Hence, though they are extremely powerful due to their simplicity, RVE models do not directly simulate the LW delay line. So, another model can be proposed, which consists of the simulation of an XZ-plane cut on the sensor, as presented in Figure 5.

### 3.2. 2.5D Model

The 2.5D model of the LW device is presented in Figure 6. The proposed model is limited to a slice of one wavelength wide, but takes into account the full thickness and length of the fabricated LW sensor. Periodic conditions are applied on the side faces and low reflection boundary conditions are imposed on the incident faces of the model. Similarly to the RVE model, the quartz substrate is divided into two physical domains to save meshing. The air medium is modeled using a single volume with electrostatic physics, while the liquid medium is modeled using the same strategy as that of the RVE LW cell, with two superposed volumes with an overall thickness of 3.5 mm. The first liquid volume has a thickness of 1 μm with high-resolution meshing, it is placed directly on the surface of the guiding layer, both electro-static and mechanical physics are considered for simulation. The second liquid volume placed over the first one is 3.5 mm thick like in experimental conditions and takes into account only the electrostatic physics behavior. To generate the LW transduction, a voltage is applied directly to the input IDT, with real thickness modeled by including the three metallic layers constituting the real electrodes (titanium, gold and titanium). Finally, the output voltage is estimated at the output IDTs, and the remaining fingers (the other half of the output IDTs) are connected to the ground plane, which is also connected to the bottom face of the quartz substrate in the model. The meshing rules are similar to the ones applied on the RVE model and are presented in Figure 7. However, a larger maximum element size was used for second liquid volume (100 μm) to reduce computing time.

Then, by estimating both the output voltage and input current, the transfer function between input ($V_{in}$) and output ($V_{out}$) voltages, and also the input impedance of the device, can be extracted. The LW sensor being inherently symmetric, the impedances of the input and output IDTs are equal.

Geometrically, according to its reduced dimensions, our 2.5D model corresponds to 1/39th of the real experimental LW device (IDT aperture on Y axis). Thus, the presented impedance of the real experimental device is similar to the 39 FEM model disposed in the parallel circuit. Subsequently, the FEM model impedance is 39 times greater than the real one. Finally, we noticed that the technological fabrication process of the Love Wave sensor induces an electrical experimental “default” in the transducer that we consider as a purely parasitic resistance ($R_{paras}$) in series with the simulated impedance. Hence, it is easy to recombine the S-parameter model from the estimated input impedance by using Equations (Equation 1) and (Equation 2) based on Figure 1: (1)$$Z_{11}=Z_{22}=\frac{Z_{FEM}}{39}+R_{paras}$$
(2)$$S_{21}=S_{12}=\frac{Z_{11}}{Z_{11}+50}\cdot \frac{50}{Z_{22}+50}\cdot \frac{V_{out}}{V_{in}}$$
with:$Z_{11}$: calculated input impedance;$Z_{22}$: calculated output impedance;$Z_{FEM}$: estimated impedance by the 2.5D model.

Unlike the RVE strategy, the 2.5D model can simulate the entire device power flow; hence, it is the most powerful tool for simulating the LW sensor in this paper. However, the computation time is quite long, approximately 6 min per frequency. For improved performance, this work also proposes a method to reduce computational time. Figure 8 and Figure 9 show, respectively, the simulation of the Shear Horizontal mechanical deformation at the liquid–guiding layer interface and the voltage distribution given by the 2.5D LW model.

### 3.3. Used Love Wave Device and Liquid Samples

In this work, we used an LW device with a 40 μm wavelength (λ in Figure 1), with IDTs consisting of 88 doubled fingers and a 164.25λ long acoustic path. Experiments are carried out with three different LW sensors, the main difference being the guiding layer thickness of 3.7, 4.8 and 6.3 μm ± 0.1 μm. The protocol used in this study is similar to that applied in previous work [9]. We used an open-loop characterization with a Vector Network Analyzer (VNA, Copper mountain Plannar 304/1), directly connected to the LW sensor with pogo-pins of a custom-made test cell. A PDMS chip is used to localize a liquid sample on the sensor surface, including over the IDTs, contrary to the more common method based on a microfluidic chip localizing the solution on the acoustic path only, as described in [22,23]. Finally, the selected aqueous sample solutions are drop-casted with a micro-pipette on the LW sensor surface and VNA characterizations at low and high frequencies are carried out to measure capacitive and electroacoustic responses. The injection of 100 μL liquid samples is carried out via the drop-casting method using a micro-pipette. After every liquid test, the Love Wave device is cleaned with ethanol, acetone and deionized water, followed by a drying step with nitrogen. Four delay lines of each batch were tested with 15 different solutions of aqueous ethanol, aqueous glycerol, aqueous Polyethylene Glycol (PEG600) and mixtures of PBS and water.

## 4. Results

### 4.1. RVE Simulation of Love Wave Device Electrical Responses

As the LW RVE model corresponds to 1/39th of the IDT aperture and 1/44th of the total number of IDT fingers, the calculated impedance given by the model can be used to estimate the sensor impedance by a simple division of 39 × 44, with the addition of the supposedly parasitic resistivity. However, this calculation does not take into account the influence of the feedlines, which is hypothesized to be small at the frequency of interest.

The RVE model estimated impedance of the sensor in air is presented in Figure 10. As expected, the capacitance is constant below 100 MHz, as measured in [9]. Resistivity, however, exhibits two small peaks, the first one below 10 MHz and the second around the resonant frequency $f_{r}$, and is almost negligible at other frequencies. This means that the hypothesis of the measured resistivity being assimilated to the parasitic impedance can be validated. These peaks can be associated with an acoustic transduction, which supports the choice of using a fixed frequency of 50 MHz to estimate the “pure” dielectric response of the LW sensor given by the model.

Figure 11 presents the estimated and measured capacitance at 50 MHz for both measurement and FEM estimations with an adjacent liquid medium. On the basis of the previous results, the dielectric permittivity of the $SiO_{2}$ guiding layer is chosen as a fitting parameter with a value of 6.2. As expected, a thicker guiding layer improves the trapping of the electric field, increasing the capacitance value. The estimation is close to the measurement results.

Figure 12 shows the comparison of measurements and RVE estimations for the liquid dielectric permittivity variation at 50 MHz for different guiding layer thicknesses, taking the response to air as a reference. Results are close, although perfectible, they clearly exhibit the same behavior according to the liquid medium dielectric permittivity as described in [24]. As expected, a thicker $SiO_{2}$ guiding layer limits interaction of the electric field with the liquid medium, decreasing the sensitivity of the LW sensor to the liquid dielectric permittivity. The same fitting parameters are taken into account for the 2.5D model approach in the next part.

### 4.2. 2.5D Model: Estimation of Love Wave Device Electrical Responses

As presented in Figure 2, the electrical response of the sensor is estimated from the LW reflected energy. Thus, the 2.5D model can use only the LW input IDTs to estimate the $S_{11}$ parameters, instead of a full XZ-cut of the model, minimizing computational time.

As for the LW RVE model, experimental measurements and 2.5D FEM estimation of the LW sensor impedance and equivalent capacitance according to the thickness of the guiding layer were investigated. In addition, the capacitance of the LW sensor according to the dielectric permittivity of the liquid medium, with the response in air taken as a reference, was estimated and compared with the experimental results and the RVE model. We noticed that, with a silicon dioxide dielectric permittivity of 6.2, the 2.5D model gives results very similar to the RVE estimation represented in Figure 11 and Figure 12; thus, they are also in good accordance with the experimental results. However, the 2.5D electrical response of the LW device uses more computer memory and takes around 20 times longer to compute.

### 4.3. Discussions on the Love Wave Dielectric Sensing Estimation

The similarity of the responses obtained with both models and the experimental results presented in Figure 11 and Figure 12, show that both the 2.5D and RVE LW sensor response estimations are promising for electrical sensing, though better fitting could be achieved. Indeed, the current measurement protocol seems to produce high discrepancies, making any fitting approximate. There is a greater variation in the capacitance of the LW sensor for a lower thickness of the guiding layer, as observed in [24]. A thinner guiding layer or greater dielectric permittivity of the guiding layer might allow for improved capacitive sensing of the LW sensor, but with the major drawback of having less energy for the acoustic transduction. The presented sensors show less electrical sensitivity for aqueous solutions with a high dielectric constant around 80 (cf. Figure 12). This could mean that for a small variation around this point, an almost pure acoustic sensing of the LW device can be achieved, independently of the permittivity. The higher electrical sensitivity of LW sensor with respect to the liquid dielectric permittivity is achieved for 1<ϵ<40, which is typically the dielectric constant of oil and organic-composed liquids. The electrical behavior of the LW sensor being correctly modeled, the next section will present the results of the 2.5D and RVE models for the LW device acoustic and electro-acoustic sensing estimation.

## 5. Results: Love Wave Acoustic and Electro-Acoustic Sensing Estimation

### 5.1. RVE Model-Based Estimation of Love Wave Device Acoustic Sensing Responses

The eigenfrequency solver gives a complex frequency as output, where the real part corresponds to the angular frequency and the imaginary part to the damping of the mode. In this work, the complex property of the eigenfrequency will not be investigated, as the real part already carries the change in LW celerity, which allows one to estimate the resonance frequency of the system.

Figure 13 shows the LW sensor resonance frequency for variations in the thickness of the guiding layer for the air medium. The estimation fits almost perfectly with the measurement and the computed confidence interval caused by the variation in the thickness of the measured guiding layer (≈0.1 μm) can explain the discrepancies. Although the results are computed extremely quickly, less than 20 s per point, it gives no information about the impedance of the LW sensor and the transiting energy.

In Figure 14, we plot the relative changes in the resonance frequency of the LW sensor according to the variations in the liquid mechanical parameters. The graph is linearized by calculating $p=\sqrt{\rho \eta }\;\left[{\mathrm{kg}\cdot m}^{-2}{\cdot s}^{-1/2}\right]$, a parameter defined in [11,25]. ρ and η are, respectively, the density and dynamic viscosity of the liquid medium. The linearity observed is in accordance with the literature, with an experimental slope of around $-850\;\mathrm{ppm}\cdot {\mathrm{mL}}^{-1/2}\cdot {\mathrm{cP}}^{1/2}\cdot g^{1/2}$ [26]. However, we can observe a large confidence interval of experimental results in Figure 14 which indicates discrepancies in the $f_{r}$ measurement method. Furthermore, we can observe a poor concordance between experiments and the estimated LW sensor response to liquid mechanical parameters.

A first response to this observation is that it is needed to take into account the multiphysical aspect of the experimental LW device response (i.e., both electrical and acoustic behavior) and choose the reference measurements with great attention. As the RVE model does not correctly simulate the impedance changes around the resonance frequency due to the geometric simplifications, it is not possible to simultaneously estimate the influence of both physics. Section 4.1 showed that, for variation of the electrical permittivity of the liquid medium around 80, the capacitive response of the sensor shows a limited influence. Hence, by changing the reference of the LW sensor response measurement from air to deionized (DI) water, small variations of the liquid medium electrical permittivity would have a limited influence on the input impedance.

Moreover, we chose to convert the phase response measurement into time delay (flight time). Indeed, the $S_{21}$ phase response showed greater repeatability and reduced disparity of measurements compared to the relative frequency shift.

To compare both the measurements and the model, the resonance frequency simulations can be converted to an equivalent LW time delay with Equation (Equation 3):(3)$$\Delta t=LAP(1/f_{r}-1/f_{r_{ref}})$$
with LAP being the number of wavelengths in the length of the acoustic path, from the middle of the input IDT to the middle of the output IDT (in our case, LAP = 208.25).

The results in Figure 15 highlight that both RVE estimation and measurement are extremely close when changing the reference from air to DI water. This confirms the low incidence of small variations around high dielectric permittivity in the sensor response.

### 5.2. 2.5D Model-Based Estimation of Love Wave Electro-Acoustic Sensing Responses

#### 5.2.1. 2.5D Model Simulation Results

The frequency signature of the LW sensor by using 2.5D simulations in air is presented, respectively, in Figure 16 for the $S_{11}$ parameter and in Figure 17 for the $S_{21}$ parameter. In this first step, the entire LW device 2.5D model described in Section 3.2 is used, which needs more computation time. Figure 16b and Figure 17b show similar graphs based on 12 sensor responses, measured three times. We can notice that all 2.5D estimations are in accordance with measurements, although perfectible. It can be noted that the samples with a 3.7 μm thickness guiding layer have a greater parasitic resistance (≈180 Ω), subsequently considerably modifying the $S_{11}$ parameter compared to other sensors, which was already measured in [9].

#### 5.2.2. Reducing the Computation Time with LW Sensor 2.5D Model

According to the description of the LW 2.5D model presented in Section 3.2, three responses associated with three different types of liquid media, taking into account mechanical and/or electrical parameters, are presented in Figure 18. First, the blue and red curves show the estimated response of the 2.5D LW sensor model by taking into account the typical 2.5 mm thick liquid layer. The blue curve estimates the LW sensor 2.5D model response according to the variation of the liquid viscosity only, simulating a purely mechanical phenomenon. The red curve presents the LW sensor 2.5D model response according to the experimental electrical and mechanical parameters of the concentration of the aqueous glycerol solution, which has an increase in the liquid viscosity for a decrease in its electrical permittivity. As the electric coupling from the input to output IDT occurs due to the presence of the liquid (as shown on power flow in Figure 2), estimations need to be time-gated to separate electrical coupling from the acoustic energy, as explained in detail in previous work [9,27,28]. Stable results are obtained with 100 points per frequency scan, which requires 10 h long simulation per liquid. For both those estimations, a frequency sweep of 10 MHz around the resonance frequency is conducted.

In order to reduce the computation time of the 2.5D simulations for LW electroacoustic sensing, we proposed to minimize the liquid layer thickness of the original 2.5D model. This “derived model” is named the “reduced model” in Figure 18. We can notice on the green curve of Figure 18, by decreasing the thickness of the liquid from 2.5 mm to 50 μm in the reduced model, that the estimated response of the LW sensor to liquid mechanical and dielectric parameters remains in good agreement with the two models simulating the liquid real thickness and the permittivity variations. The “reduced model” does not need time-gating and subsequently the simulation of one frequency carries all the information needed for LW electro-acoustic sensing in aqueous solutions while reducing the computation time to 10 min.

Figure 19a,b present the response $S_{21}$, both the estimation from the 2.5D reduced model of the LW device and the measurement, according to variations of the liquid mechanical parameters at the resonance frequency. The Love Wave response reference was taken in air. These simulations also take into account the liquid electrical parameters in accordance with the glycerol proportion in the aqueous solution. Figure 20 presents the $S_{11}$ response of the LW sensor according to the same parameters. It can be observed that the simulations give similar results as measurements. Slight differences may be caused by the approximate fitting of the $S_{11}$ parameter of the LW sensor, which did not affect the RVE estimation.

### 5.3. Discussion about the Electro-Acoustic Sensing Estimation

Both RVE and 2.5D models give similar results for the acoustic response of the sensor. With the RVE model, the electro-acoustic response could not be simulated, whereas the 2.5D model can give dielectric, acoustic and electro-acoustic estimation but with the drawback of enormous computational resource requirements. Nonetheless, in cases of limited dielectric variation, a pure acoustic response can be hypothesized and the LW RVE model gives results extremely close to experiments.

A strategy of simplification is proposed, such as decreasing the liquid layer thickness to produce an effect similar to the “time gating” method already demonstrated in [9]. The simplification reduced the computational time by a factor 100. However, this method also suppresses a powerful tool in the estimation of the dielectric constant of the liquid. Indeed, the electrical coupling from input to output IDT can be used to estimate a greater liquid dielectric constant compared to the LW sensor input capacitance [9]. The use of separate LW sensor models, simulating each branch of the power flow separately (presented in Figure 2), might allow us to achieve more accurate estimations and even faster simulations.

Table 1 synthesizes the performances of both the model according to the variation of a liquid dielectric and mechanical properties, calculation time and calculation resources needed.

## 6. Conclusions and Further Work

This work paves the way to a new method of using the SAW sensor in a liquid medium. Some of our previous work [9] showed that, experimentally, both electrostatic and electroacoustic mechanisms influence the SAW sensor response. A simple numerical analysis and a novel protocol enhanced Love Wave sensor potential in liquid media. Those previous results proved that both phenomena are intrinsically linked, but did not propose a concrete means to separate their contribution in the overall response of the sensor. The new models proposed in this work allow one to validate the possibility of decorrelation of each physical phenomenon. Moreover, the understanding of each component of the global response of the sensor is greatly improved. Implementing specific models could allow the detection of biological targets with specifically designed sensors. For instance, the presented models could be used to design a specific geometry to enhance the capacitive sensing ability of a Love Wave device, making it efficient for Electrochemical Impedance Spectroscopy (EIS) according to different guiding layer materials which would also be used as a dielectric sensing layer, without reducing the acoustic sensitivities. This new design would be multiphysically optimized using the model for a selected bio-chemical target, improving the sensing performances. Extracellular polymeric substance (EPS) layers, isolated from a Tunisian thermophilic microalga strain *Graesiella*, are currently being investigated as promising candidates for such an application with heavy metals [29], but have yet to be included into a SAW device which would be optimized for both sensing mechanisms. Also, the model could be used to determine and suppress variability in the response of the sensors, such as parasitic resistivity, to normalize the sensitive response despite technological discrepancies.

Further work will be carried out on measurement protocols with a test cell better adapted to improve the accuracy of dielectric measurements. There is good scope for improving the models to estimate the input impedance with better accuracy and reduced computational time.

## Figures and Tables

**Figure 1 sensors-24-02976-f001:**
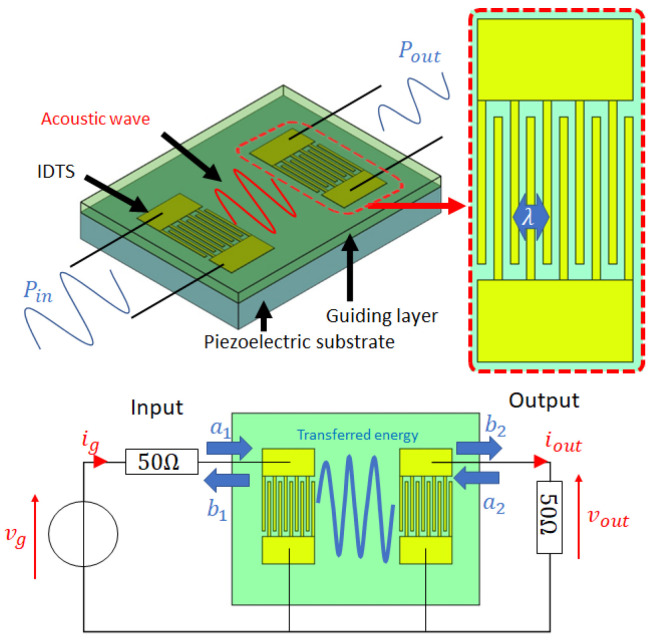
Simplified LW sensor structure (**top**) and basic read-out instrumentation (**bottom**). According to S parameters convention, $a_{i}$ and $b_{i}$ are, respectively, incident and reflected waves.

**Figure 2 sensors-24-02976-f002:**
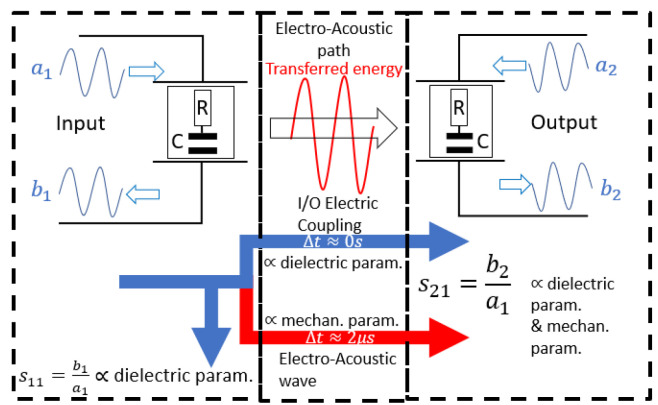
Schematic of the LW sensor power flow ($a_{i}$: incident wave; $b_{i}$: reflected wave; Δt: propagation time).

**Figure 3 sensors-24-02976-f003:**
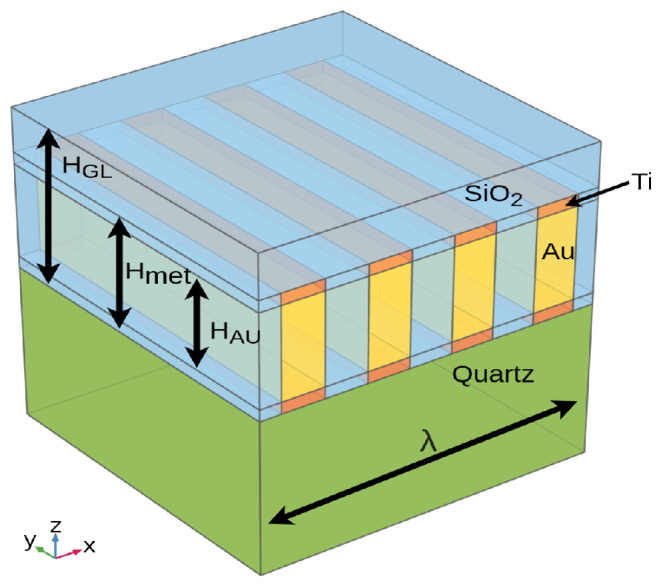
SAW structure typical RVE model.

**Figure 4 sensors-24-02976-f004:**
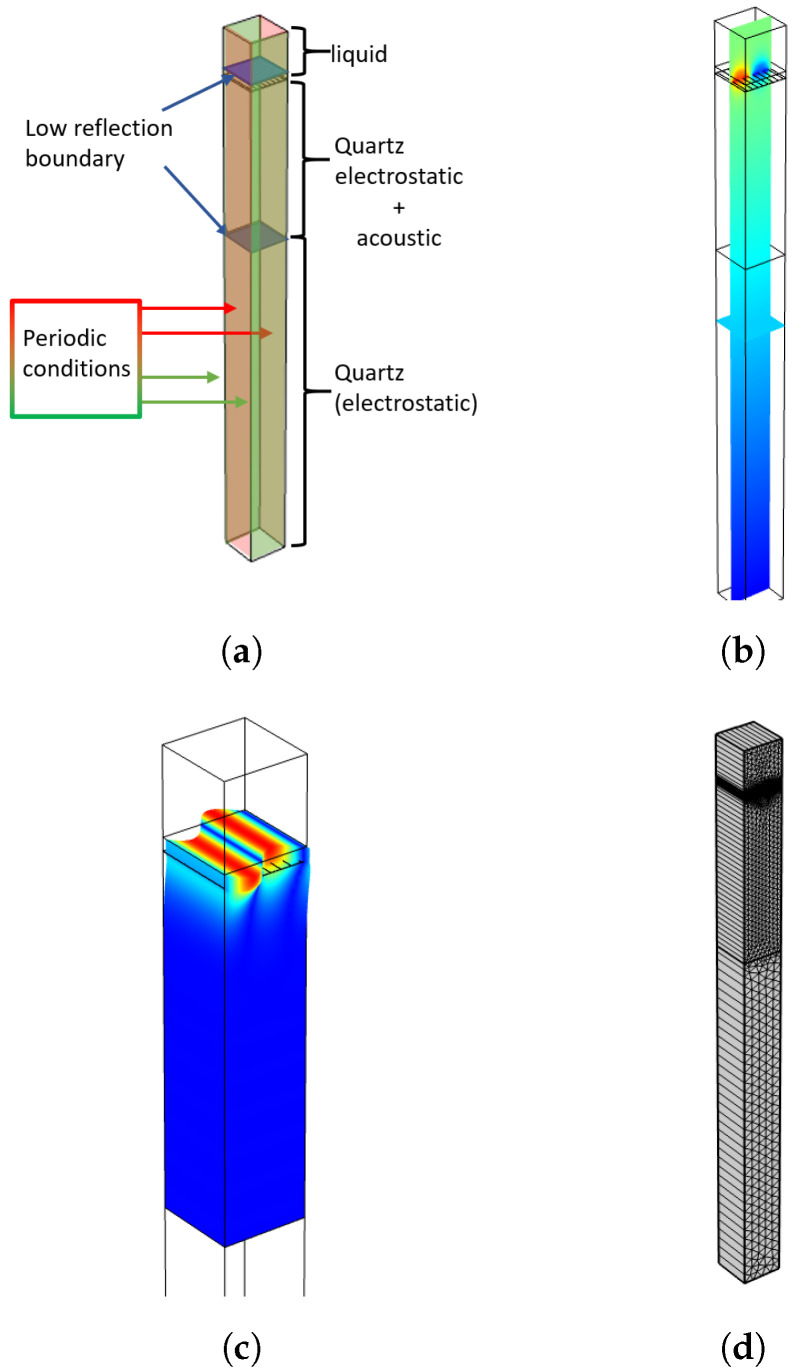
Love Wave RVE model and simulated qualitative results. (**a**) LW sensor body representation; (**b**) Applied voltage distribution (excitation signal); (**c**) Mechanical displacement simulation; (**d**) Proposed mesh for simulation.

**Figure 5 sensors-24-02976-f005:**
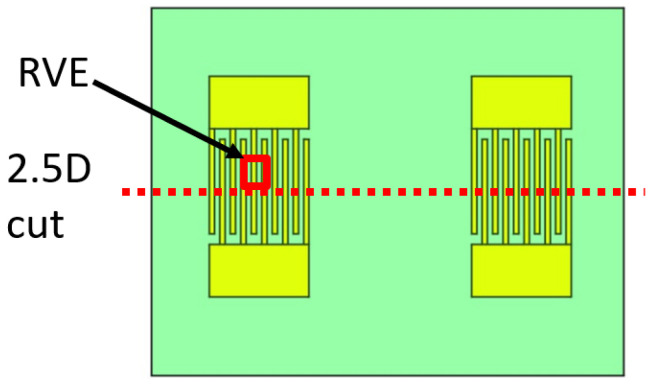
Location of the RVE and 2.5D models on the LW device top view.

**Figure 6 sensors-24-02976-f006:**
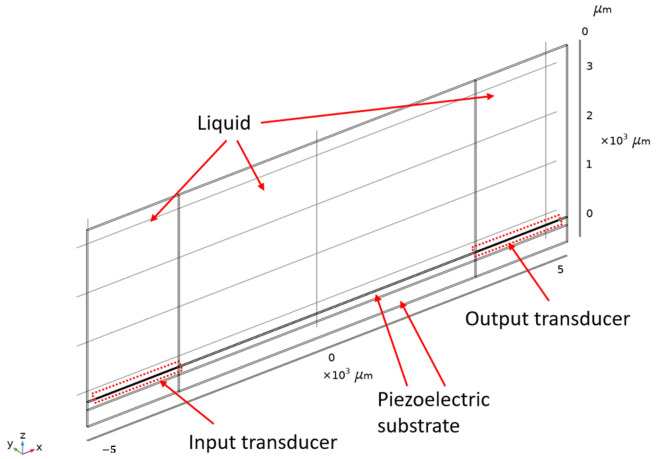
Love Wave sensor with liquid sample: 2.5D modeling.

**Figure 7 sensors-24-02976-f007:**
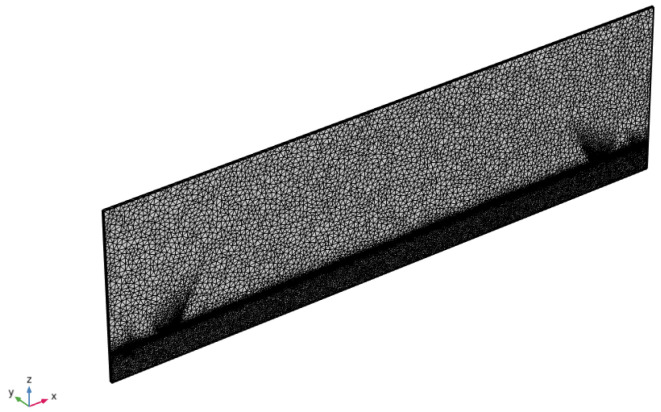
Proposed mesh for 2.5D model.

**Figure 8 sensors-24-02976-f008:**
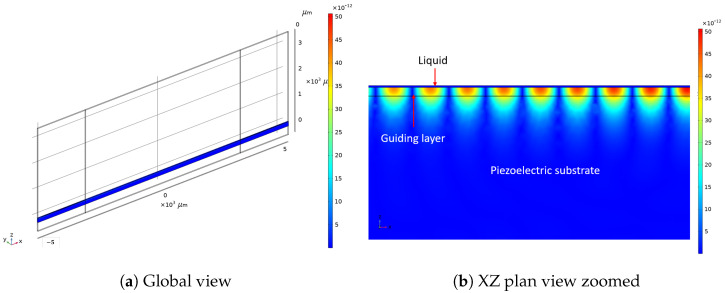
The 2.5D LW model simulation of mechanical deformation distribution (colorbar: amplitude of deformation in meters).

**Figure 9 sensors-24-02976-f009:**
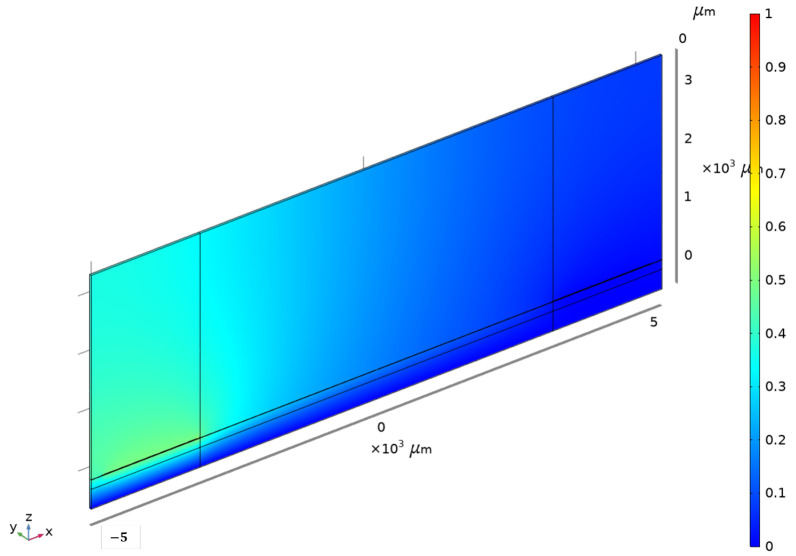
Voltage distribution in LW 2.5D model (colorbar: voltage distribution in volts).

**Figure 10 sensors-24-02976-f010:**
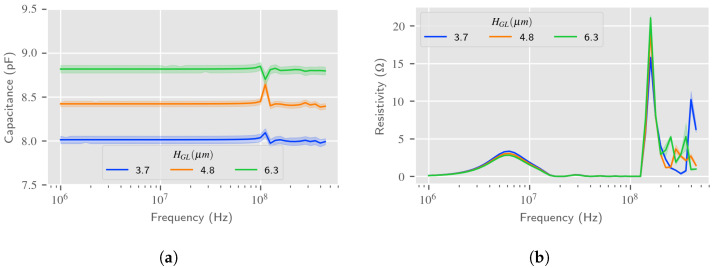
FEM estimation of the LW input IDT impedance using RVE model in air. (**a**) Capacitance vs. frequency—RVE estimation; (**b**) Resistivity vs. frequency—RVE estimation.

**Figure 11 sensors-24-02976-f011:**
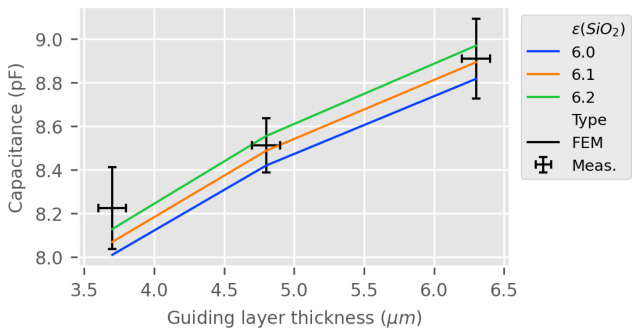
LW IDT capacitance according to the $SiO_{2}$ guiding layer thickness at 50 MHz in air—RVE estimation with 3 values of $SiO_{2}$ permittivity vs. measurements.

**Figure 12 sensors-24-02976-f012:**
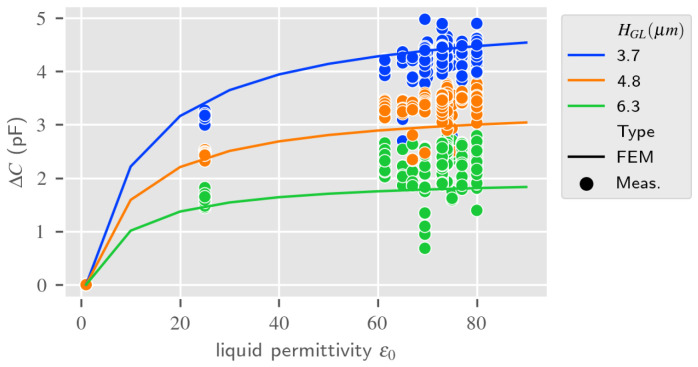
IDT capacitance according to the liquid electrical permittivity at 50 MHz (liquid electrical conductivity = 0)—RVE estimation vs. measurements.

**Figure 13 sensors-24-02976-f013:**
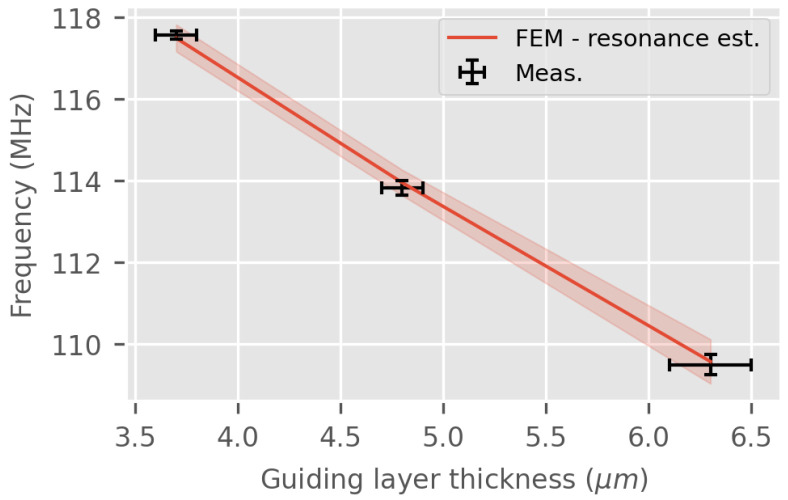
LW resonance frequency $f_{r}$ according to the guiding layer thickness in air—FEM RVE estimation and experimental measurements (confidence interval estimated using typical guiding layer thickness variation ≈0.1 μm).

**Figure 14 sensors-24-02976-f014:**
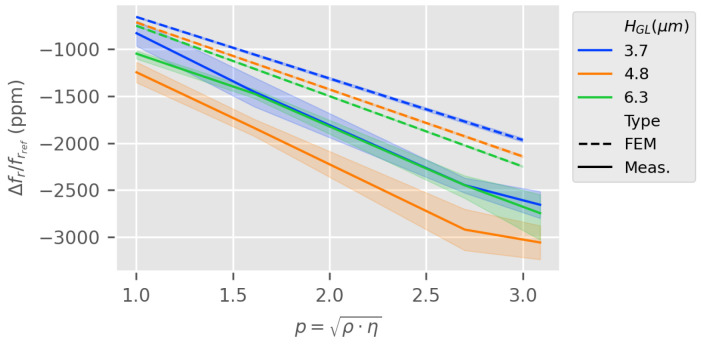
Relative resonance frequency shift according to the liquid medium mechanical parameters (reference being LW sensor resonance frequency in air—$f_{r_{ref}}$—RVE estimation vs. measurement.

**Figure 15 sensors-24-02976-f015:**
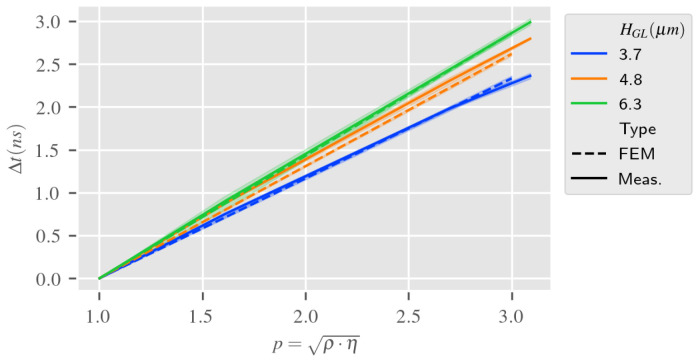
Time delay (input to output IDTs) according to liquid mechanical parameters per guiding layer thickness $H_{GL}$, RVE estimation vs. measurement (reference is DI water).

**Figure 16 sensors-24-02976-f016:**
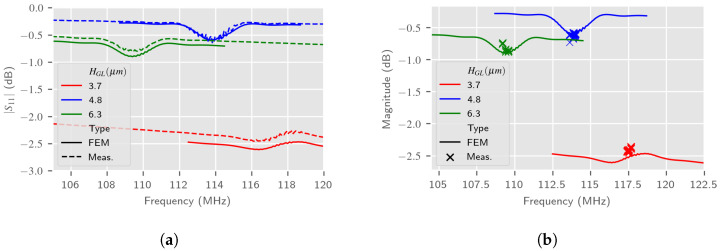
Comparison of estimated LW sensor $S_{11}$ spectrum in air with 2.5D simulations and experimental measurements according to the guiding layer thickness $H_{GL}$. (**a**) $S_{11}$ typical spectrum; (**b**) Experimental $f_{r}$ and estimated $S_{11}$ spectrum.

**Figure 17 sensors-24-02976-f017:**
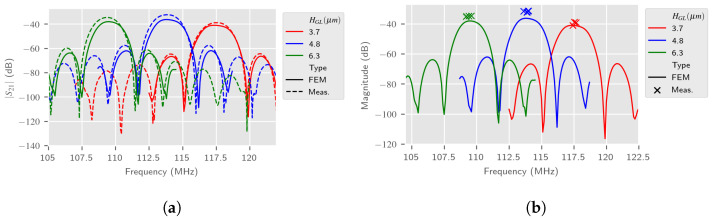
Comparison of estimated LW sensor $S_{21}$ spectrum in air with 2.5D simulations and experimental measurements according to the guiding layer thickness $H_{GL}$. (**a**) $S_{21}$ typical spectrum; (**b**) Experimental $f_{r}$ and estimated $S_{21}$ spectrum.

**Figure 18 sensors-24-02976-f018:**
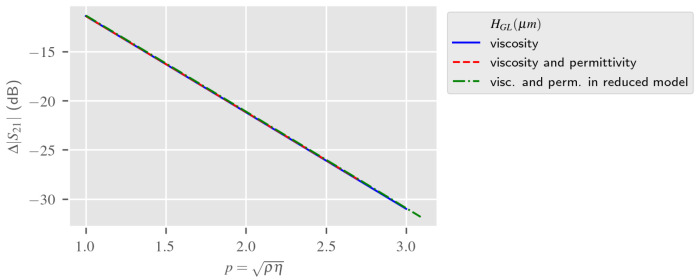
Liquid mechanical and electrical parameters’ influence on LW sensor with full and reduced liquid thickness on $S_{21}\left(@f_{r}\right)$—2.5D Estimation.

**Figure 19 sensors-24-02976-f019:**
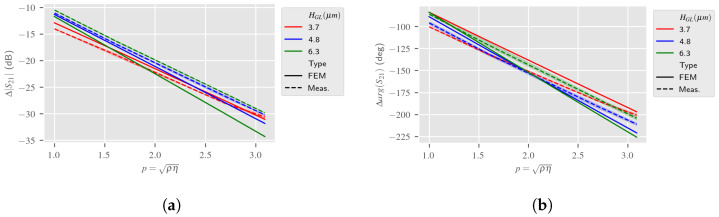
Impact of liquid mechanical parameters on LW sensor $S_{21}$ at $f_{r}$. Simulated and experimental responses according to the guiding layer thickness $H_{GL}$—2.5D reduced model (reference: LW response in air). (**a**) Magnitude of $S_{21}$ (at $f_{r}$); (**b**) Phase of $S_{21}$ (at $f_{r}$).

**Figure 20 sensors-24-02976-f020:**
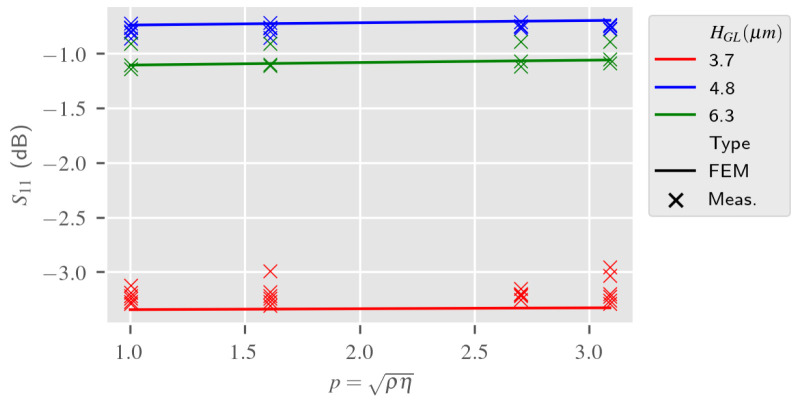
Liquid mechanical parameter impact on $S_{11}$ at $f_{r}$ simulated and experimental responses according to the guiding layer thickness $H_{GL}$—2.5D reduced model (reference: LW response in air).

**Table 1 sensors-24-02976-t001:** Synthesis of the results according to models.

	RVE	2.5D
Liquid dielectric properties simulation only (ϵ)	Possible for $f<f_{r}$	Yes, for all frequencies (tested from 1 MHz up to 1 GHz)
Liquid mechanical properties simulation only (ρ, ν)	Resonance frequency $f_{r}$	All parameters of interest (S-parameters)
Liquid electro-mechanical (dielectric and mechanical) properties estimation (ρ, ν, ϵ)	Simulation of one mechanism at a time	Simulation of both mechanisms
Calculation time	20 s/point	10 min/point
Calculation resources needed	Light (RAM < 8 Go)	Heavy (RAM > 64 Go)
Accuracy with measurements for dielectric properties	Good	Good
Accuracy with measurements for mechanical properties	Very good	Good and upgradable
Accuracy with measurements for dielectric and mechanical properties		Good and upgradable

## Data Availability

Dataset available on request from the authors.

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
