# Peer review of "FEM Modeling Strategies: Application to Mechanical and Dielectric Sensitivities of Love Wave Devices in Liquid Medium"

_sensors, 2024, doi:10.3390/s24102976_

Round 1

Reviewer 1 Report

Comments and Suggestions for Authors

Thank you for the interesting and comprehensive paper. However, there are several questions and suggestions from my side.

1. In the introduction part authors state that use of FEM for electroacoustic sensors "remains sparse or even unexplored". I can not agree with this idea. A quick search performed in Google scholar gives 126 000 results for "Surface acoustic wave FEM analysis", 17 700 of which are published in 2020 or later. 

Next authors state that these computations require "enormous computational resources". This statement is only half true. Everything depends on the scale of the task. Several examples:

1) Fu, C., Elmarzia, O., & Lee, K. (2014, March). Implementation of wireless battery-free microgyrosensor by utilizing one-port SAW delay line and an antenna with double resonant frequencies. In Reliability, Packaging, Testing, and Characterization of MOEMS/MEMS, Nanodevices, and Nanomaterials XIII (Vol. 8975, p. 897503). SPIE.  - Very good paper, describing a full-3D model of a SAW device. Simulation is made in ABAQUS. 

2) Li, B., Zhang, Q., Zhao, X., Zhi, S., Qiu, L., Fu, S., & Wang, W. (2022). A General FEM Model for Analysis of Third-Order Nonlinearity in RF Surface Acoustic Wave Devices Based on Perturbation Theory. Micromachines13(7), 1116. - COMSOL is used to perform the simulation of an RVE of SAW reflectors.

3)  Lukyanov D. et al. Design and research of bidirectional surface acoustic wave delay line fabricated using laser ablation method //Fiber Lasers and Glass Photonics: Materials through Applications. – International Society for Optics and Photonics, 2018. – Т. 10683. – С. 106833O. - an aproach very similar to 2.5D mentioned in the paper is presented.

4) Lukyanov D. P., Kukaev A. S., Shevchenko S. Y. Design optimization of a microgyrocsope on standing surface acoustic waves //Integrated Navigation Systems (ICINS), 2017 24th Saint Petersburg International Conference on. – IEEE, 2017. – С. 1-3. - also shows a full 3D FEM model built in OOFELIE::Multiphysics.

Some of mentioned researches were performed using a typical office computer. Some minor limitations were introduced to reduce the model size, but, nevertheless, meaningful results were obtained even for full-3D models.

2. Figures 1 and 2 represent two options of the sensor, which is described in text. But what does this sensore actually sense? Also, it is worth adding some comments for the design, shown in Fig. 2. 

3. Authors give a lot of details about the model design, computational resources and estimated, but donot provide any information about the mesh they used. So, it is not possible to judge about the real computational size of the model and its dimensional precision. 

4. Both in RVE and 2.5D cases authors limit some dimensions to 1 wavelegth. Hence, in the frequency domain the wavelength is swept in some interval, are these dimensions adequately adopted during the computation?  

Comments on the Quality of English Language

Several possible misprints:

1) "remained carried out" in line 31

2) "multiphsiycal" in line 95

3) ITD in line 207

4) "cf." in line 297

Author Response

Dear Reviewer

I am writing to express my sincere gratitude for your time and effort in reviewing our article.

Your insightful comments and constructive feedback have been immensely valuable in improving the quality and clarity of our work. We appreciate the thoroughness with which you approached the review process, and your insights have undoubtedly contributed to the overall enhancement of the manuscript.

We have carefully considered each of your suggestions and have made the necessary revisions to address the concerns raised. Your remarks are presented in the rest of this letter in bold, while our responses are put in plain text.

Thank you for the interesting and comprehensive paper. However, there are several questions and suggestions from my side.

  1. In the introduction part authors state that use of FEM for electroacoustic sensors "remains sparse or even unexplored". I can not agree with this idea. A quick search performed in Google scholar gives 126 000 results for "Surface acoustic wave FEM analysis", 17 700 of which are published in 2020 or later. 

Next authors state that these computations require "enormous computational resources". This statement is only half true. Everything depends on the scale of the task. Several examples:

1) Fu, C., Elmarzia, O., & Lee, K. (2014, March). Implementation of wireless battery-free microgyrosensor by utilizing one-port SAW delay line and an antenna with double resonant frequencies. In Reliability, Packaging, Testing, and Characterization of MOEMS/MEMS, Nanodevices, and Nanomaterials XIII (Vol. 8975, p. 897503). SPIE.  - Very good paper, describing a full-3D model of a SAW device. Simulation is made in ABAQUS. 

2) Li, B., Zhang, Q., Zhao, X., Zhi, S., Qiu, L., Fu, S., & Wang, W. (2022). A General FEM Model for Analysis of Third-Order Nonlinearity in RF Surface Acoustic Wave Devices Based on Perturbation Theory. Micromachines13(7), 1116. - COMSOL is used to perform the simulation of an RVE of SAW reflectors.

3)  Lukyanov D. et al. Design and research of bidirectional surface acoustic wave delay line fabricated using laser ablation method //Fiber Lasers and Glass Photonics: Materials through Applications. – International Society for Optics and Photonics, 2018. – Т. 10683. – С. 106833O. - an aproach very similar to 2.5D mentioned in the paper is presented.

4) Lukyanov D. P., Kukaev A. S., Shevchenko S. Y. Design optimization of a microgyrocsope on standing surface acoustic waves //Integrated Navigation Systems (ICINS), 2017 24th Saint Petersburg International Conference on. – IEEE, 2017. – С. 1-3. - also shows a full 3D FEM model built in OOFELIE::Multiphysics.

Some of mentioned researches were performed using a typical office computer. Some minor limitations were introduced to reduce the model size, but, nevertheless, meaningful results were obtained even for full-3D models.

Response: Both introduction and a few paragraphs were updated to accommodate the recent work done on this topic. The authors mainly meant to say that articles which focus on FEM for SAW are not as numerous as their applications. We agree of the poor choice of words and toned down some of the sentences while adding a few of the proposed citations. However, to our knowledge, no paper tried to simulate the entire Love Waves device, IDTs included, in liquid medium, while taking account both electrostatic and mechanical behavior of the transducer as well as the liquid.

  1. Figures 1 and 2 represent two options of the sensor, which is described in text. But what does this sensore actually sense? Also, it is worth adding some comments for the design, shown in Fig. 2. 

Response: A short paragraph was added to simplify the common applications of SAW sensors. (lines 66-69, 79, 83)

  1. Authors give a lot of details about the model design, computational resources and estimated, but donot provide any information about the mesh they used. So, it is not possible to judge about the real computational size of the model and its dimensional precision. 

Response: Information and figures regarding mesh were added to section 2: “Material and methods”.

  1. Both in RVE and 2.5D cases authors limit some dimensions to 1 wavelegth. Hence, in the frequency domain the wavelength is swept in some interval, are these dimensions adequately adopted during the computation?  

Response: The dielectric sensitivity is estimated at a lower frequency than the acoustic resonance for both models. Thus, because no acoustic generation is created far from the resonance frequency, the dimensions do not affect the response of the sensor.

For the electro-acoustic sensing part of the work, the RVE model only estimates the resonant frequency, which is linked to the wavelength imposed by the IDTS geometry (distance between é fingers at the same potential – λ in Fig. 1), hence there is no need to adapt the dimensions for the computation. The 2.5D model generates the acoustic wave around the resonant frequency, and the meshing is sufficient to allow the acoustic wave to have enough elements to be correctly modelized. However, no effects of the OZ dimension (device width) were shown to affect the response according to change in the wavelength. This can be explained by the use of periodic conditions on both OXZ plane and the directivity of the acoustic wave toward the OY direction.

Reviewer 2 Report

Comments and Suggestions for Authors

In the article, the authors have considered the theoretical study of a biosensor that allows to determine both the mechanical and electrical properties of a liquid medium using Love waves (LW). The theoretical approach is based on the experimental data obtained earlier. The characteristics of a biosensor using two calculation methods as RVE and 2.5D model by the Comsol software have been investigated. The satisfactory accordance between the experimental and experimental data has been obtained.

There are two mistakes in the text:

(1) Line 91. Should be written "...multiphysical ..."

(2) Line 319. No definition of the p quantity.

I should do two remarks in a way of the manuscript improvement.

(1) To improve the perception of the article, I should recommend that the authors make a summary table containing the key calculated and experimental data. In the manuscript, the conventional sensitivity parameters are absent. As a consequence, there no comparison of the considered sensor with another types of acoustoelectronic sensors.

(2) The Conclusion section is written in general words. There should be added the concrete proposals about the advantage of a chosen sensor scheme as well as its practical application possibility.

Author Response

Dear Reviewer

I am writing to express my sincere gratitude for your time and effort in reviewing our article.

Your insightful comments and constructive feedback have been immensely valuable in improving the quality and clarity of our work. We appreciate the thoroughness with which you approached the review process, and your insights have undoubtedly contributed to the overall enhancement of the manuscript.

We have carefully considered each of your suggestions and have made the necessary revisions to address the concerns raised. Your remarks are presented in the rest of this letter in bold, while our responses are put in plain text.

In the article, the authors have considered the theoretical study of a biosensor that allows to determine both the mechanical and electrical properties of a liquid medium using Love waves (LW). The theoretical approach is based on the experimental data obtained earlier. The characteristics of a biosensor using two calculation methods as RVE and 2.5D model by the Comsol software have been investigated. The satisfactory accordance between the experimental and experimental data has been obtained.

There are two mistakes in the text:

(1) Line 91. Should be written "...multiphysical ..."

(2) Line 319. No definition of the p quantity.

Response: The first error was corrected, and citations are rearranged to better point out the definition of the square root ratio of the viscosity and density, which we chose to call “p” for simplification (line 332-333).

I should do two remarks in a way of the manuscript improvement.

(1) To improve the perception of the article, I should recommend that the authors make a summary table containing the key calculated and experimental data. In the manuscript, the conventional sensitivity parameters are absent. As a consequence, there no comparison of the considered sensor with another types of acoustoelectronic sensors.

Response: A table was added to the last part of the article as a summary for both models, according to both physical mechanisms and calculation results & performances. We added the typical sensitivity slope constant to “p” as a comparison factor to other kind of acoustoelectronic sensors (line 335).

(2) The Conclusion section is written in general words. There should be added the concrete proposals about the advantage of a chosen sensor scheme as well as its practical application possibility.

Response: A concrete proposal is now added to the conclusion, and currently investigated material based on extracellular polymeric substances (EPSs) is introduced which appears as a fitting candidate for both mechanical and dielectric sensing, but it is yet to be included with a sensor that would be optimized for both mechanisms.

Round 2

Reviewer 1 Report

Comments and Suggestions for Authors

Thank you for tour abnswers and corrections. The paper became much clearer. 

Please, take care about the following:

1) line 132 has a trouble with citation.

2) references 6 and 4, 19 and 23 are the same. 

Finally, you say that "a few of proposed references were added", but I don't see any. Of course, the choice of materials to cite is up to the authors, so, plese, concider this as just a negligible remark. 

Author Response

Thank you again for your review.

Thank you for tour abnswers and corrections. The paper became much clearer. 

Please, take care about the following:

1) line 132 has a trouble with citation.

2) references 6 and 4, 19 and 23 are the same. 

Finally, you say that "a few of proposed references were added", but I don't see any. Of course, the choice of materials to cite is up to the authors, so, plese, concider this as just a negligible remark. 

There was an error with our reference file, which was corrected. This correction added citation line 132 (Li et al. 2022)